# Development of an Integrated EEG/fNIRS Brain Function Monitoring System

**DOI:** 10.3390/s21227703

**Published:** 2021-11-19

**Authors:** Manal Mohamed, Eunjung Jo, Nourelhuda Mohamed, Minhee Kim, Jeong-dae Yun, Jae Gwan Kim

**Affiliations:** 1Biomedical Science and Engineering Department, Gwangju Institute of Science and Technology, Gwangju 61005, Korea; manalalnosh@gm.gist.ac.kr (M.M.); jeju1993do@gm.gist.ac.kr (E.J.); nonoalhodaali@gm.gist.ac.kr (N.M.); minhee@gm.gist.ac.kr (M.K.); 2N.CER Co., Ltd., Gwangju 61005, Korea; yjdea2@gmail.com

**Keywords:** electroencephalogram, functional near-infrared spectroscopy, Analog Front-End, Brain Monitoring System

## Abstract

In this study, a fully integrated electroencephalogram/functional near-infrared spectroscopy (EEG/fNIRS) brain monitoring system was designed to fulfill the demand for a miniaturized, light-weight, low-power-consumption, and low-cost brain monitoring system as a potential tool with which to screen for brain diseases. The system is based on the ADS1298IPAG Analog Front-End (AFE) and can simultaneously acquire two-channel EEG signals with a sampling rate of 250 SPS and six-channel fNIRS signals with a sampling rate of 8 SPS. AFE is controlled by Teensy 3.2 and powered by a lithium polymer battery connected to two protection circuits and regulators. The acquired EEG and fNIRS signals are monitored and stored using a Graphical User Interface (GUI). The system was evaluated by implementing several tests to verify its ability to simultaneously acquire EEG and fNIRS signals. The implemented system can acquire EEG and fNIRS signals with a CMRR of −115 dB, power consumption of 0.75 mW/ch, system weight of 70.5 g, probe weight of 3.1 g, and a total cost of USD 130. The results proved that this system can be qualified as a low-cost, light-weight, low-power-consumption, and fully integrated EEG/fNIRS brain monitoring system.

## 1. Introduction

During the last decade, there has been a noticeable improvement in the quality of life, which has led to an increase in aging among world populations [1]. In 1990, the average lifespan was 65.6 years, but later this became a serious problem as the average lifespan of 73 years was reached in 2017 [2]. This problem became worse due to the increase in people aging with poor health, and the World Health Organization reported that the percentage of diseased older people will keep increasing with time [3]. There are many age-related diseases, and Mild Cognitive Impairment, Parkinson’s disease, and Alzheimer’s Disease make up a high proportion of cases (about 51.3%) [4].

A huge number of functional brain signals are required for the better detection of these diseases; these can be recorded using many tools such as electroencephalography (EEG), functional near-infrared spectroscopy (fNIRS), and functional magnetic resonance imaging (fMRI). Currently, multimodal systems that are capable of acquiring and analyzing two or more brain signals are used to provide a comprehensive picture of brain function. In particular, a more complete picture can be gained by combining two methods that represent two different physiological process, such as neuronal electrical activity and the hemodynamic activity of the brain.

EEG is the most commonly used method for assessing the electrical activity of the brain by noninvasively recording brain signals using electrodes placed on the subject’s scalp. The major components of the EEG system are electrodes, which are used for sensing EEG signals; an amplification circuit, which is used to amplify the EEG signals; analog to digital converters (ADCs), which are used to digitize the EEG signals; and a recording system, which is used to display and store the EEG signals. At present, most EEG systems are based on Analog Front-Ends (AFEs), which add integration to the system. EEG has many advantages, such as its low-cost, portability, high time resolution, and ability to carry out long-term monitoring, and fMRI and fNIRS are the most common methods used for assessing the hemodynamic activity of the brain. fNIRS has many advantages over fMRI, such as its low-cost, portability, and noninvasive nature. The main components of fNIRS systems are sources (such as LEDs, laser diodes, and VCSELs) with a driving circuit, detectors such as photodiodes (PDs) and silicon photomultipliers (SiMP) for detecting light, amplifiers, filters to reduce noise, and a recording system to display and store the fNIRS signal. The advantages of EEG and fNIRS make them the most suitable techniques for multimodal studies [5]. 

Clinically, it has been proven that integrated EEG/fNIRS systems have the ability to distinguish people with Parkinson’s disease [6]; additionally, some studies have been carried out on fNIRS, demonstrating the potential of fNIRS to study brain functional connectivity in neurodegenerative diseases such as MCI [7,8]. 

In the literature, several combined EEG and fNIRS systems have been described. In 2013, an integrated EEG/fNIRS system [9] was designed; the EEG part of this system consists of 12 active wet Ag/AgCl electrodes placed on the subject’s scalp using the 10–10 EEG electrode positioning standard, and the fNIRS part consists of eight dual-wavelength LEDs (760 nm and 850 nm) to act as sources and four silicon diodes to act as detectors, resulting in 32 fNIRS channels with a source–detector separation (SDS) of about 20 to 63 mm. The combination of the two parts results in a discrete component-based EEG/fNIRS system with separate ADCs for acquiring EEG and fNIRS signals; an EEG and fNIRS recording resolution of about 16 bits; and a sampling rate of 1024 SPS and 8 SPS for EEG and fNIRS recordings, respectively. This system has many advantages, such as its light weight (about 90 g) and high number of EEG and fNIRS channels. However, it also has many disadvantages, such as its large size and the fact that it consumes a high power of about 400 mW, because it uses separate ADCs for acquiring EEG and fNIRS signals. Additionally, this system needs extensive cabling to connect the EEG/fNIRS head element to the control unit.

In the same year, another EEG/fNIRS integrated system was proposed [10]. This system is a discrete component-based EEG/fNIRS system that has eight EEG channels and 32 fNIRS channels. The fNIRS channels are a combination of eight dual-wavelength LEDs (735 and 850 nm) to act as sources and eight avalanche photodiodes (APDs) to act as detectors, with a fixed SDS of about 31 mm. This system uses a shared 16-bit ADC to provide EEG and fNIRS signal recordings with sampling rates of about 320 and 20 SPS, respectively. Compared to the previous system, this system has the advantage of using a shared ADC and more sensitive detectors, but it has the problem of requiring extensive cabling to connect the EEG/fNIRS head element to the control unit, a high power consumption (about 2200 mW), and a high system weight (800 g).

In 2018, Kassab et al. [11] attempted to upgrade this system with more EEG channels (about 32), more sources (32 LEDs), and more detectors (32 APDs), but this resulted in an integrated system with increased size, weight (850 g), power consumption (2600 mW), and cabling compared to the previous system.

In 2019, Lee et al. designed an integrated EEG/fNIRS system [12] that consists of an EEG cap, fNIRS probe, EEG and fNIRS acquisition circuit, and power supply circuit. The EEG cap consists of 18 dry EEG electrodes, resulting in 16 EEG channels, and the fNIRS probe consists of two dual-wavelength LEDs (730 and 850 nm) to act as sources and six silicon photodiodes to act as detectors, resulting in eight fNIRS channels with a fixed SDS of about 27 mm. The main components of the EEG and fNIRS acquisition circuit are three ADCs, two of which are used for acquiring the EEG signal (ADS1299), with 24-bit resolution and a sampling rate of about 250 SPS, while the third ADC (ADS8688A) is responsible for acquiring the fNIRS signal, with 16-bit resolution and a sampling rate of about 5 SPS. This system has a smaller control unit size (about 70 × 70 mm^2^ × 2) and a lower power consumption (of about 18.8 mW per channel) compared to the previous systems, but still did not solve the problem of the extensive cabling required to connect the fNIRS probe and EEG cap to the acquisition circuit. 

In 2011, another approach for integrating EEG and fNIRS was implemented [13]. This approach integrates EEG and fNIRS into a microchip using custom-designed CMOS integrated circuits. This system consists of six LEDs with wavelengths of about 735 and 890 nm and 12 detectors with a 14.14 mm SDS, resulting in 24 channels for recording fNIRS signal with a 10-bit resolution and a sampling rate of about 1 SPS. The EEG signal was recorded with a resolution of about 10 bits and a sampling rate of about 128 SPS. To fabricate the microchip, a UMC 65 nm CMOS was used to result in a system with a power consumption of about 3.6 mW for the chip only. Although this system has a high number of fNIRS channels, it uses a fixed SDS, and information about the EEG channel is not provided.

In 2017, Ha, U. et al. [14] proposed the use of an integrated EEG/fNIRS system for anesthesia depth monitoring. This system consists of one detector and one dual-wavelength VCSEL, with wavelengths of about 670 and 850 nm, resulting in one channel used for recording the fNIRS signal with a 12-bit resolution and a sampling rate ranging from 20 to 80 SPS. For EEG signal recording, four active dry EEG electrodes are used, resulting in two channels for recording EEG with a 12-bit resolution and a sampling rate of about 2000 SPS. The overall system has a weight of less than 26 g and a power consumption of about 25.2 mW, which are very low values compared to all previous systems, but it has a limited number of EEG and fNIRS channels.

One year later, Xu, J. et al. [15] proposed an integrated EEG/fNIRS/electrical impedance tomography (EIT) system based on CMOS. The system consists of three blocks—an EEG block, an fNIRS block, and an EIT block—which are integrated into a microchip. The EEG block consists of dry electrodes (the number of these electrodes is not specified in the study), resulting in one channel for recording the EEG signal with a 15-bit resolution, while the fNIRS block consists of two LEDs with wavelengths of about 735 and 850 nm that act as sources and two SiPM that act as detectors, with an SDS of about 30 mm, resulting in four channels for recording fNIRS with a 12-bit resolution and a sampling rate ranging from 2 to 512 SPS. An important issue with this system is that SiPM has a very high voltage bias (about 30 V), which should be considered when mounting it on the subject’s scalp, but this system has a very low power consumption for the chip (about 0.665 mW), which is considered as an advantage of this system above all other systems. 

Considering all the advantages and disadvantages of the above systems, this study aimed to design a cost-efficient, lightweight, and-low-power-consumption integrated EEG/fNIRS brain monitoring system based on ADS1298IPAG AFE to fulfill the demand for miniaturized, light-weight, low-power-consumption, and low-cost brain monitoring systems as a potential tool to screen for brain diseases. 

## 2. System Design

In this section, the main components of the integrated EEG/fNIRS system are described.

### 2.1. Analog Front-End for EEG and fNIRS Measurements

Bio-potential signals, such as EEG and fNIRS, are considered weak signals as they have small amplitudes and therefore can be easily affected by noise. The use of an Analog Front-End (AFE) ensures the high-precision measurement of these signals due to its high resolution, high data rate, and high Common Mode Rejection Ratio (CMRR). AFE has a high level of integration, which ensures the development of medical instruments with a low cost and reduced size and power [16,17].

The proposed design is based on the ADS1298IPAG AFE IC for both EEG and fNIRS measurements. This IC is a low-power eight-channel AFE with eight high-resolution 24-bit delta-sigma (ΔΣ) Analog-to-Digital Converters (ADCs) and eight low-noise built-in Programmable Gain Amplifiers (PGAs). This AFE ensures a 0.75 mW power consumption at each channel with a data rate ranging from 250 SPS to 32 KSPS and a CMRR of about −115 dB [17]. ADS1298IPAG AFE is designed for use in ECG, EEG, and EMG measurements but can be still used for acquiring fNIRS signals [18].

An important and interesting feature that ADSA1298IPAG AFE can support is the Right Leg Drive (RLD) circuit, which was specially developed for ECG measurements but can be used for EEG and fNIRS measurements to improve the CMRR while maintaining good isolation for the power grid [19,20]. ADS1298IPAG AFE is best suited to this design due to its high CMRR, high SNR, and low power consumption compared to other AFEs [21]. 

### 2.2. Microcontroller

The Teensy 3.2 microcontroller is the main processing unit of the proposed EEG/fNIRS system. It runs at a 72 MHz clock frequency and has a 256 kB flash memory and 64 kB RAM. The DMA feature of the Teensy 3.2 makes it suitable for the proposed system as it can acquire, compress, and transfer data in real time. 

### 2.3. Power Supply and Protection Circuit

Power is supplied to the EEG/fNIRS system by a 2000 mAh Lithium Polymer (Li-Polymer) battery. These batteries cannot handle both over and under voltages and can catch fire when abused. Therefore, a protection circuit that is directly connected to the battery is required (Li-Polymer batteries have a built-in protection circuit). Additionally, two additional protection circuits are added; the first circuit is based on BQ29700DSET, which is a voltage and current protection integrated circuit for Li-Polymer batteries that is capable of disconnecting the battery charger from the processing circuit when an overcharge or overcurrent event take place, whereas the second circuit is based on BQ24380DSGT, which is an overvoltage and overcurrent protection IC and Li+ charger front-end protection IC that is capable of disconnecting the Li-Polymer charge management controller (MCP73811/2) from an external power source when an overvoltage or overcurrent event take place.

EEG/fNIRS systems are considered noise-sensitive systems; therefore, highly regulated output voltages with a low noise spectral density are required. This can be achieved using TPS72301DBVT and TPS73201DBVT low-dropout regulators.

## 3. Implementation

The proposed system consists of an EEG/fNIRS acquisition circuit, EEG/fNIRS power supply circuit, and a personal computer (PC). Figure 1 shows a general block diagram of the EEG/fNIRS system.

### 3.1. EEG/fNIRS Acquisition Circuit

The EEG/fNIRS acquisition circuit consisted of an EEG/fNIRS probe, an LED Switching circuit, a PD amplification circuit, and an EEG/fNIRS measurement and control circuit.

#### 3.1.1. EEG/fNIRS Probe

The EEG/fNIRS probe used was a flexible probe that consisted of LEDs and PDs for fNIRS signal sensing and Ag/AgCl electrodes for EEG signal sensing. The flexibility of the probe enabled its light weight and tight attachment to the forehead.

Two 3030 bi-color LEDs (Shenzhen Fedy Technology Co., Ltd., China) were used. These LEDs are capable of emitting near-infrared light at two different wavelengths (735 and 850 nm). They had a power dissipation of about 200 mW and a Luminous intensity of about 40–50 mW at a 735 nm wavelength and about 25–40 Mw at an 850 nm wavelength. Important features of these LEDs are their light weight and small size (3.21 mm × 2.35 mm).

The light emitted from these LEDs passes through the skull in a banana-shape and is then sensed by five OE-PD0064-T PDs (Opto ENG Co., Ltd., Korea). These PDs are high-speed Si PIN PDs that respond to wavelengths ranging from 700 to 1100 nm and have a peak wavelength of about 970 nm and a size of about 4 mm × 5.4 mm. The distance between neighboring PDs, PD1 and LED1, PD3 and LED1, PD3 and LED2, and PD5 and LED2 is 30 mm, and the distance between PD2 and LED1 and PD4 and LED2 is 5 mm, as shown in Figure 2.

The use of two LEDs and five PDs with the previously specified distances results in six fNIRS channels (two short-distance channels and four long-distance channels). Each PD senses light when the nearest LED is switched on, except for the third PD, which can sense the light from both LEDs as they are at an equal distance from them. The two short-distance channels are more sensitive to activity in the superficial layers of the subject’s head (scalp and skull); this activity can be regressed from the four long-distance channels, which are sensitive to both the brain and superficial layers of the subject’s head, resulting in them acquiring deep brain tissue activity only [22,23,24].

For sensing the EEG signal, four H124SG (Kendall, Australia) Ag/AgCl electrodes are used; these are about 24 mm in diameter and are fixed to holes in the flexible probe via 4 mm buckles. These four electrodes result in two channels for sensing EEG signal; two of these electrodes are placed on the forehead at Fp1 and Fp2, while the others are placed behind the ears at A1 and A2 to work as a reference electrode and ground.

Figure 3a shows the EEG/fNIRS flexible probe with a size of about 716 mm × 60 mm. This probe is connected to the processing unit via a micro HDMI FPC flat cable (100 cm).

#### 3.1.2. Light-Emitting Diodes (LEDs) Switching Circuit

This circuit is responsible for turning LEDs on and off. At the beginning, LED1 is turned on to emit light with a wavelength of 735 nm, then turned off. After that it is turned on again to emit light with a wavelength of 850 nm. The same process is carried out for LED2.

This circuit consists of four 2N3904 bipolar junction transistors (Fairchild Semiconductor Cor., Rectron, South Portland) that act as switches. While the transistor is switched on, a current passes through the emitter to supply the LED after it passes through a 33 Ω resistor (in the case of 735 nm) or a 100 Ω resistor (in the case of 850 nm) to limit it.

#### 3.1.3. Photodiodes (PDs) Amplification Circuit

This circuit is responsible for amplifying PD outputs (increasing the power of the PD signal). It consists of two LM324M/NOPB operational amplifiers (Texas Instruments, Dallas, TX, USA) with a feedback resistance of about 1 MΩ and a power supply of about 5 V. This circuit is also responsible for reducing the PD dark current by using a voltage divider with values of 100 and 3.3 KΩ at the positive input of the amplifier (PD0064-T PDs have a dark current of about 10 nA, which affects the fNIRS signal when the room is dark and results in noise in a form of high-amplitude peaks).

The outputs of this circuit are considered as inputs to the ADS1298IPAG AFE at its differential analog positive inputs (IN1P, IN2P, IN3P, IN4P, and IN5P), and the differential analog negative inputs of ADS1298IPAG AFE are connected to the ground.

#### 3.1.4. EEG/fNIRS Measurement and Control Circuit

This circuit consists of an ADS1298IPAG AFE (Texas Instruments, Dallas, TX, USA) and a Teensy 3.2 Arduino (Adafruit, USA) board. The ADS1298IPAG AFE is responsible for acquiring EEG and fNIRS signals. The IN1P, IN2P, IN3P, IN4P, and IN5P differential analog-positive inputs are connected to the PD amplification circuit outputs, while the IN1N, IN2N, IN3N, IN4N, and IN5N differential analog negative inputs are connected to ground to form five inputs from five PDs, which results in six fNIRS channels.

IN6P and IN7P differential analog positive inputs are connected to EEG electrodes placed on the forehead, while IN6N and IN7N differential analog negative inputs are connected to the EEG electrode behind one ear (which represents the ground). The IN8P differential analog positive input is connected to the other EEG electrode located behind the other ear (which represents the reference electrode). The IN8N differential analog negative input is not connected, allowing IN8P to act as the RLD driver to which channel 6 and channel 7 are routed.

This circuit requires three power supplies (+3.3, +2.5, and −2.5 V); the +3.3 V is supplied from the Teensy 3.2 while the +2.5 V and −2.5 V supplies are from the power supply component.

The Teensy 3.2 is responsible for controlling the ADS1298IPAG AFE, as it controls the START, RESET, and PWDN pins and can communicate with AFE via SPI communication using the CS, MOSI, MISO, and SCLK pins. The DRDY pin is an input to Arduino; when it becomes low, this indicates that the data are ready. Figure 4 shows the EEG/fNIRS measurement, the control circuit, and the communication between them.

### 3.2. EEG/fNIRS Power Supply Circuit

The power supply circuit consists of a battery, a battery charger, a charge management controller, a voltage and current protection circuit for the battery, a voltage converter, and voltage regulators.

#### 3.2.1. Rechargeable Lithium Polymer Battery

A 2000 mAh, 3.7 V, 7.4Wh Lithium Polymer (Li-Polymer) battery (SHENZHEN TAIWAN BATTERY CO., LTD, China) is used to power the circuit; it is a rechargeable battery with a JST connector and cable length of about 5 cm. It was chosen due to its small size (66 mm × 41 mm × 6.6 mm) and light weight (38 g).

#### 3.2.2. Battery Charger

An MCP73811/2 (Microchip Technology Inc., Chandler, AZ, USA) 450 mA Li-Ion Battery Charger is used in the circuit; it is a linear charge management controller that is capable of charging the battery in the shortest time possible using a specific chare algorithm. The constant voltage regulation is fixed at 4.20 V and the constant current value is selected as 85 or 450 mA.

#### 3.2.3. Overvoltage and Overcurrent Protection Circuit

The BQ24380DSGT (Texas Instruments, Dallas, TX, USA) Overvoltage and Overcurrent Protection IC is used for protecting the battery from charging circuit failure. It works like a linear regulator to maintain a 5 V output from a power source that has a maximum voltage of 30 V. Additionally, it can disconnect this source from charging the battery in the case of an overvoltage or overcurrent event.

The BQ29700DSET (Texas Instruments, Dallas, TX, USA) Voltage and Current Protection Integrated Circuit is used as a second protection step. It disconnects the battery from the circuit during high-discharge or -charge current operation. The input voltage range should be from −0.3 to 12 V.

#### 3.2.4. Voltage Converter

The LM2664M6 (Texas Instruments, Dallas, TX, USA) Switched-Capacitor Voltage Converter is used to convert a positive voltage to a negative voltage. The input voltage range is from 1.8 to 5.5 V and the output is from −1.8 to −5.5 V with a current of 40 mA. The output from this converter is used to supply the Teensy 3.2 and the amplification circuit.

#### 3.2.5. Voltage Regulators

Two voltage regulators are used; one is the TPS73201DBVT (Texas Instruments, Dallas, TX, USA) 250-mA Low-Dropout Regulator, which has an input range of about 1.7 to 5.5 V and an adjustable output ranging from 1.2 to 5.5 V. it is designed to have an output of about 2.5 V to power the AFE. The other regulator used is the TPS72301DBVT (Texas Instruments, Dallas, TX, USA) Negative Output Linear Regulator, which gives an output of about −2.5 V to supply the AFE. It has an input voltage ranging from −10 to −2.7 V and an adjusted output voltage ranging from −10 to 1.2 V. Figure 5 shows the block diagram of the overall power supply part.

The overall EEG/fNIRS system combining both the EEG/fNIRS acquisition part and the power supply part is implemented with a PCB size of about 84 mm × 62 mm, as shown in Figure 6.

### 3.3. Personal Computer (PC)

The controller is connected to the PC using a USB to enable it store EEG and fNIRS data and display the data in real-time using a Graphical User Interface (GUI). In the GUI, EEG_CH1 and EEG_CH2 show the raw EEG data, and CH1, CH2, CH3, CH4, CH5, and CH6 are used to display the fNIRS data by calculating the oxy-, deoxy-, and total hemoglobin (∆HbO, ∆HbR, and ∆HbT) and displaying them versus time.

The GUI has the ability to choose a port that enables communication between the system and PC; also, it has the ability to save the data under a chosen name. In the beginning, the port should be chosen; then, it is necessary to press the CONNECT push button and then click the START push button to begin receiving data and display the data in the corresponding channels, as shown in Figure 7.

### 3.4. EEG/fNIRS System Cost

The EEG/fNIRS system consists of items that are easy to obtain. Table 1 shows the price of each item required in the final design without the soldering cost. Four boards should be ordered, with the price of a single board being about KRW 8750 for the processing circuit and KRW 100,000 for the flexible probe.

## 4. EEG/fNIRS System Evaluation

### 4.1. Analog Front-End Evaluation

#### 4.1.1. Internal Test Signal

To validate the ADS1298IPAG AFE, test signals can be generated internally. For this, the values registered by CONFIG1 and CONFIG2 should be changed to select the output data rate, test signal frequency, test signal amplitude, and test source. Three signals can be generated at each channel according to the selected CONFIG1 and CONFIG2 values—a square wave with a frequency of 1Hz, a square wave with a frequency 2Hz, and a DC mode output to give a constant voltage at each channel—as shown in Figure 8.

#### 4.1.2. Sinusoidal Input Test Signal

Another validation test can be conducted by supplying channel 1 with a sinusoidal signal with a peak-to-peak voltage equal to 250 mV and a frequency equal to 10 Hz. The signal is acquired correctly by AFE, as shown in Figure 9.

#### 4.1.3. ECG Signal Detection

The third means to validate the AFE is by collecting the ECG signal from channel 7 while using channel 8 as RLD, resulting in three leads. These data were recorded from a healthy subject and the result is shown in Figure 10.

#### 4.1.4. EMG Signal Detection

The fourth means to validate AFE is by collecting the EMG signal from channel 7 while using channel 8 as the ground. The EMG signal was recorded from a healthy subject while they relaxed their hand and then clenched it; the result is shown in Figure 11.

#### 4.1.5. Input-Referred Noise

Input-referred noise is the noise generated internally by the AFE; it is calculated by shorting channel 1 using a CH1SET register. Figure 12 shows the noise signal obtained with a sampling rate of 250 SPS and PGA = 1; this signal was averaged for 3025 samples to result in an average input-referred noise of about 29.9 µV.

### 4.2. EEG Signal Recording Using EEG/fNIRS System

#### 4.2.1. EEG Signal Recording

The EEG signal was evaluated by using the probe as shown in Figure 3a to acquire EEG data using channel 6, channel 7, and channel 8 with a sampling rate of 250 SPS. The data were recorded at rest, as shown in Figure 13.

#### 4.2.2. EEG Signal Recording during Eye Blink

The EEG signal was recorded during eye blinking, as shown in Figure 14. Eye blinking can be noticed in the recording, which means that the ADS1298IPAG AFE is very sensitive to muscle movements during EEG signal recording.

#### 4.2.3. EEG Signal Recording during Eyes Opened and Closed

Another means to validate the EEG signal is by comparing the alpha waves recorded while the eye is opened and closed.

The probe shown in Figure 3a was placed on a healthy subject’s scalp and the EEG signal was recorded while his eyes were closed and relaxed for 1 min and while his eyes were open for 1 min. Then, the data were analyzed to calculate the power/frequency spectrum, which shows that while the eyes were closed there was increased activity in the 4 to 20 Hz region of the frequency domain, as shown in Figure 15.

### 4.3. FNIRS Signal Using EEG/fNIRS System

#### 4.3.1. fNIRS Recording Using Solid Phantoms

fNIRS signal was evaluated using two silicon-based tissue phantoms made from polydimethylsiloxane (PDMS), curing agent, TiO2, and India ink. In the near-infrared range, PDMS has a refractive index of about 1.43 (equal to tissue refractive index). TiO2 is considered as a scattering agent for setting the scattering properties of the phantom, and the India ink set the absorption properties of the phantom.

The amount of India ink used in one phantom was more than that used in the other (darker phantom contains 220 g of silicon, 24.8 g of curing agent, 0.317 g of TiO2, and 270 µL of India ink) to represent a high absorbance of light (absorption coefficient is about 0.088 mm^−1^ and reduced scattering coefficient is about 1 mm^−1^), whereas the other had less India ink (brighter phantom contains 219.7 g of silicon, 24.7 g of curing agent, 0.314 g of TiO2, and 70 µL of India ink) to represent a lower absorbance of light (absorption coefficient of about 0.02 mm^−1^ and reduced scattering coefficient of about 1 mm^−1^). The two phantoms are shown in Figure 16.

The probe shown in Figure 3a was placed on the surface of each phantom to record the value of the voltages (represents the light intensity value) versus the number of samples for 2 min. It was found that for all fNIRS channels, the brighter phantom gives higher voltage values compared to the darker phantom, which means that the reflected and scattered light intensity sensed by the PDS is high in the brighter phantom than in the darker phantom (darker phantom absorbs more light), as shown in Figure 17.

#### 4.3.2. fNIRS Recording during Arterial Occlusion Experiment

The changes in the concentration of oxy-, deoxy-, and total hemoglobin (∆HbO, ∆HbR, and ∆HbT) were obtained from six fNIRS channels by recording fNIRS data with a sampling rate of 8 SPS during an arterial occlusion experiment. The experiment was conducted as follows: During the first 60 s, the subject’s hand was at rest without any cuff occlusion, meaning that the hemodynamic responses (∆HbO, ∆HbR, and ∆HbT) converged towards the baseline. Then, the cuff was contracted rapidly within 6 s and the contraction lasted for 3 min, resulting in blocking the arterial blood flow, causing ∆HbO and ∆HbR to diverge from each other. Finally, the cuff was released gradually so that ∆HbO and ∆HbR would begin to return to converge to the baseline values.

The ∆HbO, ∆HbR, and ∆HbT were calculated based on previous studies [25,26]. Accordingly, two assumptions were made:

(1) The background absorbance of biological tissues is negligible.

(2) In human tissue, the main contribution of chromophores is confined to oxy- and deoxyhemoglobin (HbO and HbR).

Given the assumptions listed above, the change in optical density (ΔOD) at 735 and 850 nm is related to ΔhbO and ΔHbR, as follows:(1)$$\left[\begin{array}{c} {\Delta \mathrm{OD}}^{735nm}\\ {\Delta \mathrm{OD}}^{850nm} \end{array}\right]=\left[\begin{array}{cc} \varepsilon_{HbR}^{735nm} & \varepsilon_{HbO}^{735nm}\\ \varepsilon_{HbR}^{850nm} & \varepsilon_{HbO}^{850nm} \end{array}\right]\left[\begin{array}{c} ∆\mathrm{HbR}\\ ∆\mathrm{HbO} \end{array}\right]\;L$$
(2)$$\left[\begin{array}{c} {\Delta \mathrm{OD}}^{735nm}\\ {\Delta \mathrm{OD}}^{850nm} \end{array}\right]=\left[\begin{array}{cc} \varepsilon_{HbR}^{735nm} & \varepsilon_{HbO}^{735nm}\\ \varepsilon_{HbR}^{850nm} & \varepsilon_{HbO}^{850nm} \end{array}\right]\left[\begin{array}{c} ∆\mathrm{HbR}\\ ∆\mathrm{HbO} \end{array}\right]\;d.\;DPF$$
where:

${\Delta \mathrm{OD}}^{735nm}$ is the change in optical density at 735 nm;

${\Delta \mathrm{OD}}^{850nm}$ is the change in optical density at 850 nm;

$\varepsilon_{HbR}^{735nm}$ is the extinction coefficient of HbR at 735 nm;

$\varepsilon_{HbO}^{735nm}$ is the extinction coefficient of HbO at 735 nm;

$\varepsilon_{HbR}^{850nm}$ is the extinction coefficient of HbR at 850 nm;

$\varepsilon_{HbO}^{850nm}$ is the extinction coefficient of HbO at 850 nm;

∆HbR and ∆HbO are the change in the concentration of oxy- and deoxy-hemoglobin;

L is the optical path length between the light source and detector;

d is the source–detector distance;

DPF is the differential path length factor.

Then, the ∆HbO and ∆HbR values will be:(3)$$\left[\begin{array}{c} ∆\mathrm{HbR}\\ ∆\mathrm{HbO} \end{array}\right]=(\frac{1}{d.\;DPF})\left[\begin{array}{cc} \varepsilon_{HbR}^{735nm} & \varepsilon_{HbO}^{735nm}\\ \varepsilon_{HbR}^{850nm} & \varepsilon_{HbO}^{850nm} \end{array}\right]\left[\begin{array}{c} {\Delta \mathrm{OD}}^{735nm}\\ {\Delta \mathrm{OD}}^{850nm} \end{array}\right]$$
where:(4)$${\Delta \mathrm{OD}}^{735nm}=OD_{Transient}^{735nm}-OD_{baseline}^{735nm}=\log\left(I_{baseline}^{735nm}-I_{Transient}^{735nm}\right)$$
(5)$${\Delta \mathrm{OD}}^{850nm}=OD_{Transient}^{850nm}-OD_{baseline}^{850nm}=\log\left(I_{baseline}^{850nm}-I_{Transient}^{850nm}\right)$$

Finally, the change in the concentration of the total hemoglobin (∆HbT) can be obtained by:(6)∆HbT=∆HbO+∆HbR

The ∆HbO, ∆HbR, and ∆HbT values recorded during the arterial occlusion experiment carried out in one channel are shown in Figure 18. This result reveals that the proposed system can efficiently analyze the change in hemodynamic concentration.

### 4.4. Dark Noise and Dynamic Range of fNIRS Measurements

Electrical performance parameters, including dark noise and the dynamic range of fNIRS measurements, were tested according to [27]. The probe was put on the darker phantom mentioned in Section 4.3.1 and the LEDs were switched off, the phantom was covered by a black cloth to keep background light out, and the detector readings were recorded for 5 min. The mean value for these recordings was calculated to represent the dark noise, and was found to be 0.07204 V. Then, using this value, the dynamic range of fNIRS measurements was calculated and found to be 74.45 dB.

### 4.5. Electrical Crosstalk between EEG and fNIRS Signals

As the proposed system combines two electrically separate systems (EEG and fNIRS), electrical crosstalk between these systems may occur due to field coupling between the switching current in the fNIRS optode wires and the EEG electrodes [5,28,29]. To assess the impact of the LED switching current in EEG, a resistive network phantom with electrodes positioned according to 10–20 system positions was used following the method of a study carried out in 2017 by von Lühmann, et al. [28], which specified the phantom details and the experimental conditions that should be followed to assess the crosstalk in EEG in hybrid systems. In accordance with [28], a phantom consisting of a 200 Ω resistor, to simulate the electrode skin impedance, and a 2.7 KΩ resistor, to simulate electrode skin conduction, was implemented. Then, at electrode position F8, a sine-wave test signal with an amplitude of 150 µV and different frequencies was supplied to the network. These two LEDs were placed between Fp1 and F3 and Fp2 and F4; then, the test signal was recorded for 70 s while the LEDs were on or off at positions Fp1 and Fp2. To evaluate the crosstalk using MATLAB 2021a, the FFT spectrum of the last 60 s of each measurement was calculated and the spectrum was converted to Vrms amplitudes, as shown in Figure 19a,b. For the two channels Fp1 and Fp2 using different test signal conditions while the LEDs were on or off, the signal intensities were extracted at a frequency of 16 Hz, which represents the switching frequency, and then summed to give the Pnoise value, which is used to calculate the strength of the fNIRS switching noise introduced in the EEG. Figure 19c shows the results.

Comparing the results shown in Figure 19a with the findings in the study mentioned [28], we can conclude that there is no crosstalk effect in the EEG when using our proposed system, as there are no significant peaks at the switching frequency of 16 Hz or its harmonics in the FFT spectra of the acquired signal. Additionally, the strength of the fNIRS switching noise introduced in the EEG is not high (0.002), the noise of around 16 Hz has the same value while the LEDs are on or off for both Fp1 and Fp2, and there is no deviation (almost zero).

### 4.6. Mechanical Robustness of the Flexible EEG/fNIRS Probe

A mechanical test was implemented based on reference [30] to check the robustness of the flexible PCB. The flexible PCB was subjected to 200 cycles of bending at different angles. Then, CH1 and CH6 fNIRS data were recorded for about 2 min from the brighter and darker phantoms mentioned in Section 4.3.1. The data shown in Figure 20 reveals that the flexible PCB remained functional after the 200 bending cycles with a slight change in the CH1 and CH6 fNIRS data. This result indicates that the flexible PCB is robust and suitable for this application.

## 5. Discussion

A comparison between the implemented system and seven previous systems is shown in Table 2. The proposed system is based on ADS1298IPAG AFE, which consists of eight channels; five of these are used to acquire the fNIRS signal and the remainder are used to acquire the EEG signal, which means that our system uses shared ADCs for both EEG and fNIRS. This is why the system size is the smallest (about 84 × 62 mm^2^) compared to all the other systems and the lightest in weight (about 73.5 g) compared to discrete components-based systems. Another advantage that ADS1298IPAG AFE adds to the system is its low power consumption, as the system consumes only 0.75 mW for each channel (about 6 mW for the whole system), which is less than the power consumption of the reported discrete component-based systems and some microchip-based systems. Additionally, this system operates with a Common Mode Rejection Ratio (CMRR) of about −115 dB, which is better than that of other systems. Finally, the system costs only about KRW 151,180 (about USD 130).

Our proposed system uses two LEDs and five PDs with two SDS (30 mm and 5 mm), resulting in two short-distance fNIRS channels and four long-distance fNIRS channels. This adds an advantage to the system, as the two short-distance channels are more sensitive to activity in the superficial layers of the subject’s head, which can be regressed from the four long distance channels are sensitive to both the brain and superficial layers of the Subject head, resulting in acquiring deep brain tissue activity.

The synchronicity of the EEG and fNIRS signal acquisition and the use of a common ground helps to minimize noise and the electrical crosstalk of LEDs switching current into EEG, which adds two essential advantages to our proposed system compared to other systems.

The small size (about 84 × 62 mm^2^) and light weight (70.5 g) of the processing unit; the flexibility, small size (716 × 60 mm^2^) and light weight (3.1g) of the probe; and the non-extensive cabling (micro HDMI FPC flat cable is used) ensure the wearability of our proposed system while maintaining patient comfort. Additionally, the flexibility of the probe ensures its good contact with the patient’s forehead, while its stability is supported by the four wet Ag/AgCl electrodes and an adhesive tape Velcro strip. One issue with our proposed system is the limited EEG channels (only two channels) compared to the discrete component-based systems, but these two channels are considered to be supportive channels for the six fNIRS channels, which are sufficient to reach our goal of having a potential tool that could screen for brain diseases such as MCI, as proven in [7]. In addition, this system may be appropriate for people who are interested in stress studies such as anger, anxiety, etc. [31,32,33].

Additionally, a comparison between our proposed system and three commercial EEG/fNIRS systems available on the market was undertaken. One of these systems is the DSI-Hybrid-EEG+fNIR wearable sensing system [34], which consists of eight active dry EEG electrodes with four pairs of fNIR optodes arranged around them. This system has the ability to record fNIRS signal with a sampling rate of about 15 SPS, while it can record EEG signals using eight channels with a 16-bit resolution and a 300 SPS sampling rate. This system has a very high CMMR (>120 dB), but its weight is considered heavy (about 513 g). Another commercial system is NIRSPORT2 (NIRx fNIRS/EEG) [35], which consists of 8–80 sources and 8–80 detectors with multi-modal compatibility (built-in EEG) and two options for detector sensors: either a silicon photodiode (SiPD) or an avalanche photodiode (APD). The drawbacks of this system include its heavy weight (970 g), large size (162 mm × 125 mm × 60 mm), and extensive cabling. Additionally, the fNIRS/EEG Artinis package [36] is a commercial EEG/fNIRS system that has a combined EEG/NIRS head cap and optode holders that provide full head coverage (112 fNIRS channels and 128 EEG channels); however, it also suffers from extensive cabling. The advantages of our proposed system compared to these other systems are its light weight, lack of extensive cabling, high CMRR, and low cost.

In this study, a cost-efficient, lightweight, low-power-consumption, and integrated EEG/fNIRS brain function monitoring system was designed.

## 6. Conclusions

This paper presents an integrated EEG/fNIRS brain function monitoring system that is capable of recording the EEG signal using two channels and the fNIRS signal using six channels. The components of the system were selected to provide a low-cost, light-weight, and integrated EEG/fNIRS brain monitoring system while maintaining a good resolution, high CMRR, and low power consumption.

This system was evaluated to prove its ability to efficiently and correctly acquire EEG and fNIRS signals. First, ADS1298IPAG AFE was successfully tested by generating internal test signals; then, input-referred noise was calculated, giving a result of 29.9 µV, and the dark noise and the dynamic range of the fNIRS measurements were measured, resulting in 0.07204 V and 74.45 dB, respectively. Moreover, it was proven that there is no crosstalk effect of fNIRS switching current in the EEG. Additionally, the ability of the system to correctly and efficiently acquire ECG and EMG signals was proven. Finally, the ability of the system to acquire EEG and fNIRS signals was proven by analyzing the acquired EEG signal while the subjects’ eyes were opened and closed, detecting the sensitivity of the system to eye blinking during EEG recording, and monitoring the efficient response of the system to changes in hemodynamic concentration during the arterial occlusion experiment.

## Figures and Tables

**Figure 1 sensors-21-07703-f001:**
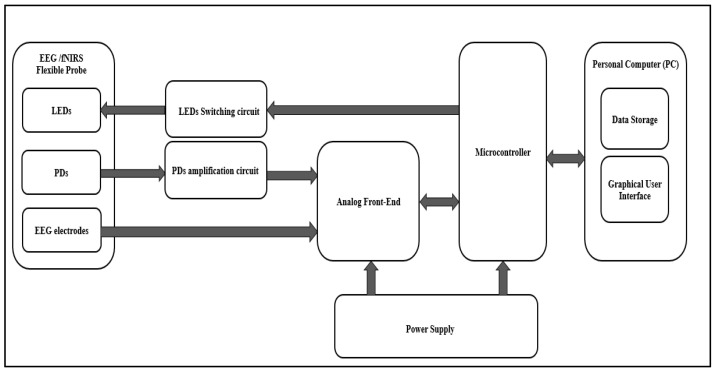
General block diagram of the EEG/fNIRS system.

**Figure 2 sensors-21-07703-f002:**
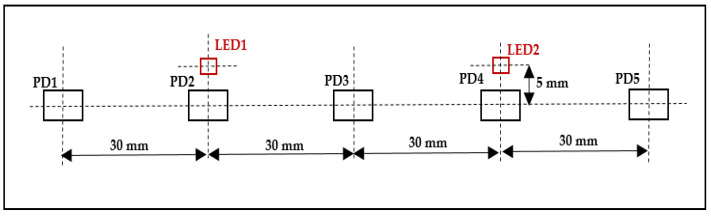
Distance between the LEDs and PDs.

**Figure 3 sensors-21-07703-f003:**
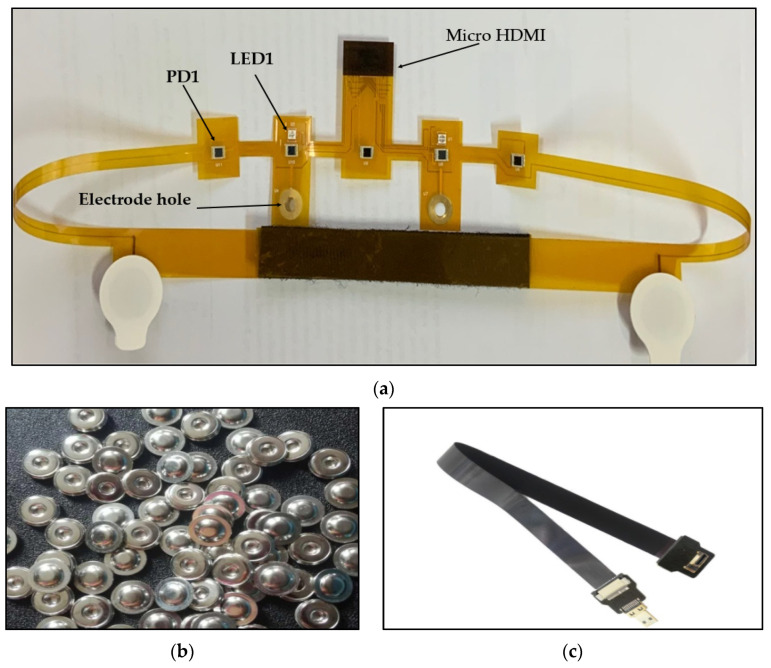
(**a**) Flexible EEG/fNIRS probe, (**b**) 4 mm buckle, (**c**) micro HDMI FPC flat cable.

**Figure 4 sensors-21-07703-f004:**
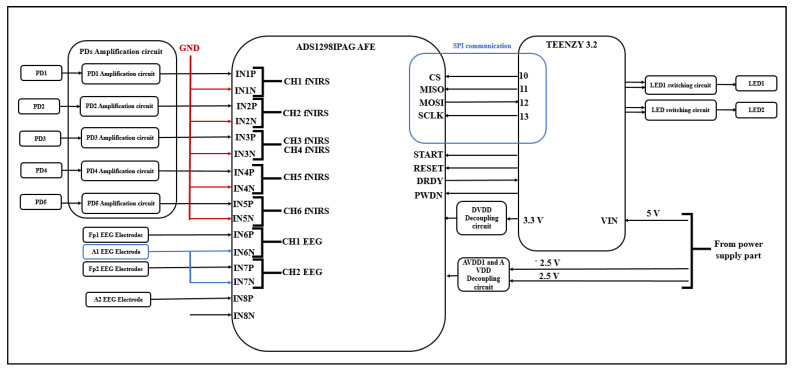
EEG/fNIRS measurement, control circuit and the communication between them.

**Figure 5 sensors-21-07703-f005:**
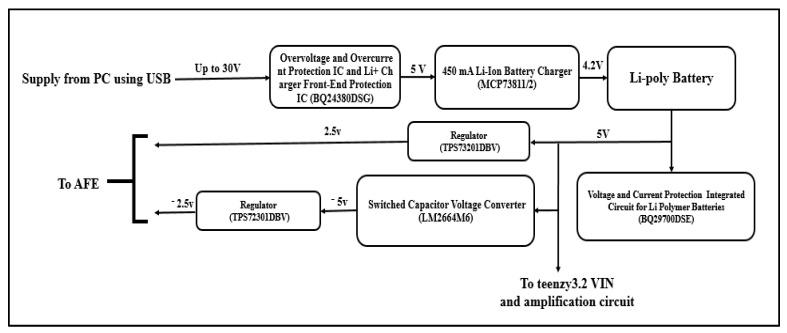
Block diagram of the overall power supply part.

**Figure 6 sensors-21-07703-f006:**
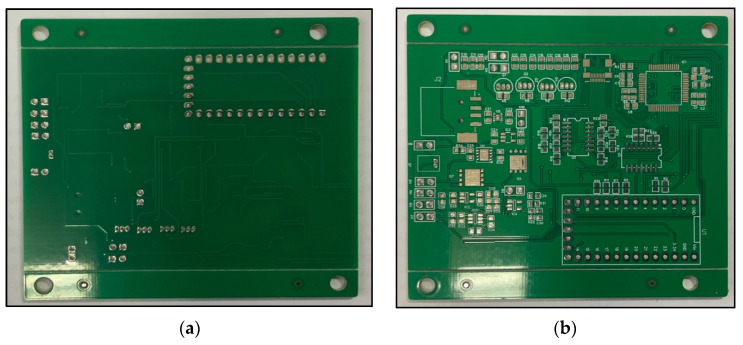
(**a**) Bottom view of the overall EEG/fNIRS system. (**b**) Top view of the overall EEG/fNIRS system. (**c**) Final EEG/fNIRS system.

**Figure 7 sensors-21-07703-f007:**
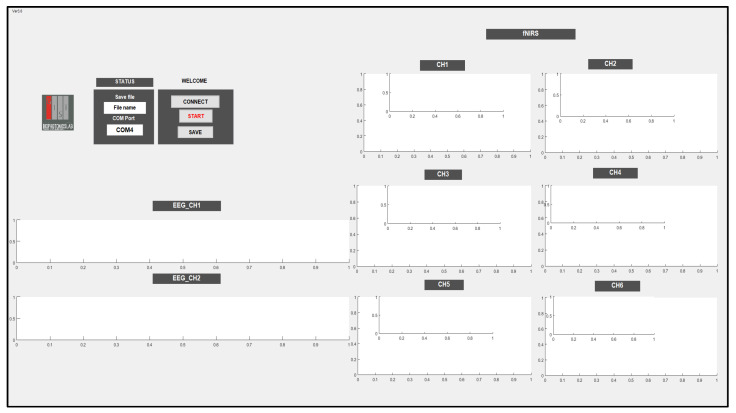
Graphical User Interface used to store and display EEG and fNIRS data.

**Figure 8 sensors-21-07703-f008:**
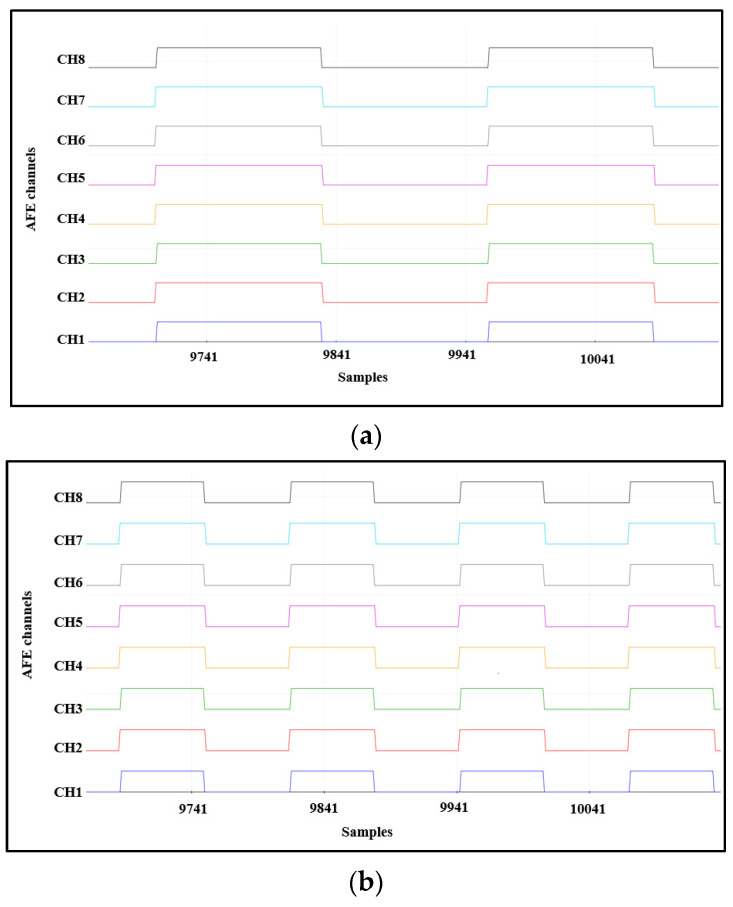
AFE internal test signals: (**a**) square wave with a frequency of 1 Hz, (**b**) square wave with a frequency of 2 Hz, and (**c**) DC mode output to give a constant voltage at each channel.

**Figure 9 sensors-21-07703-f009:**
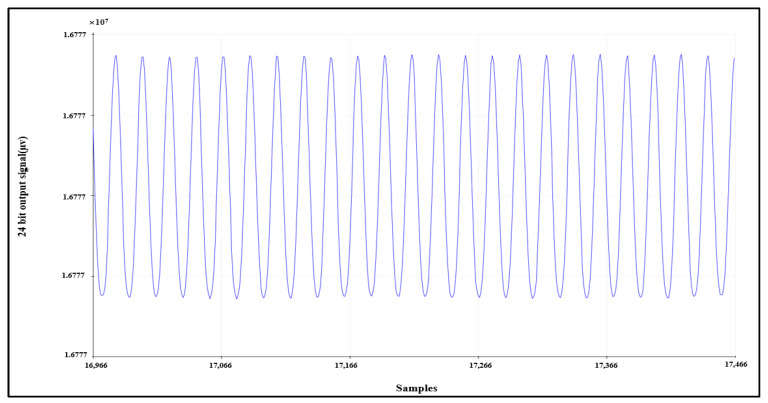
Sinusoidal input test signal.

**Figure 10 sensors-21-07703-f010:**
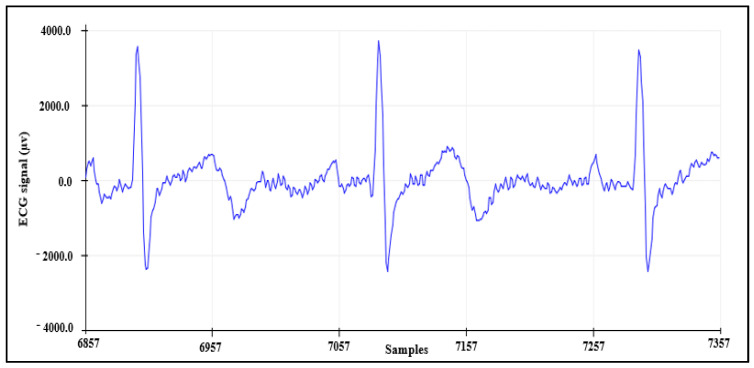
ECG signal collected using AFE.

**Figure 11 sensors-21-07703-f011:**
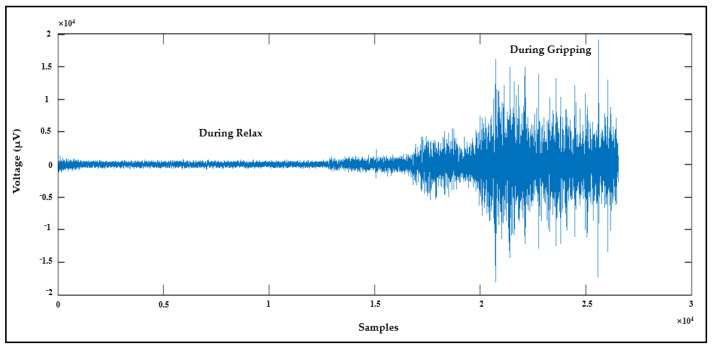
EMG signal recorded using AFE.

**Figure 12 sensors-21-07703-f012:**
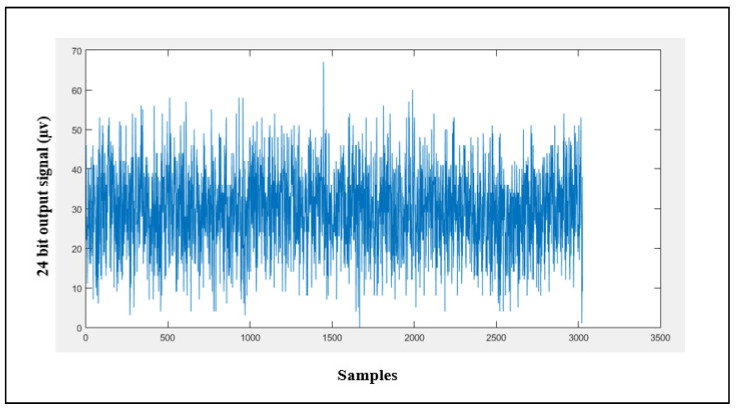
Channel 1 input-referred noise.

**Figure 13 sensors-21-07703-f013:**
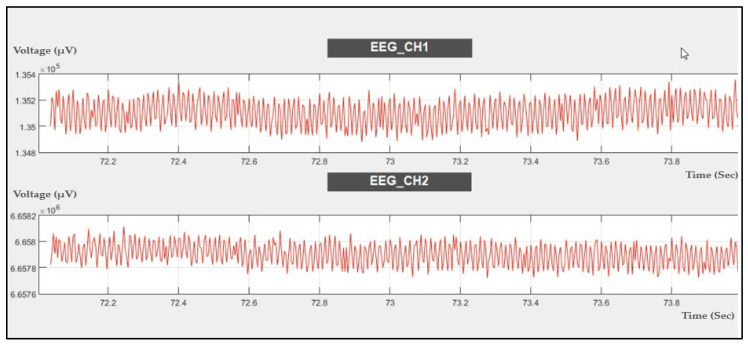
Two-channel EEG data recorded during rest.

**Figure 14 sensors-21-07703-f014:**
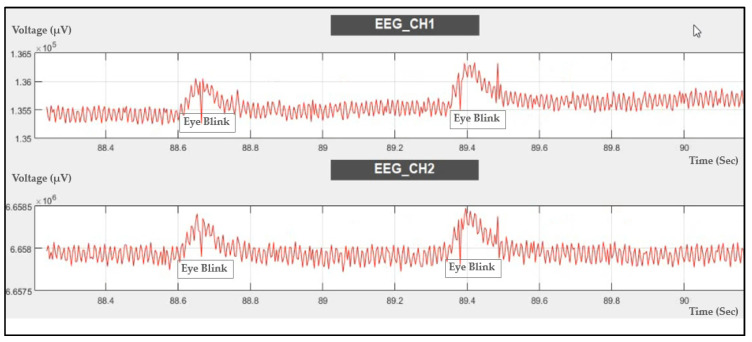
Two-channel EEG data recorded during eye blinking.

**Figure 15 sensors-21-07703-f015:**
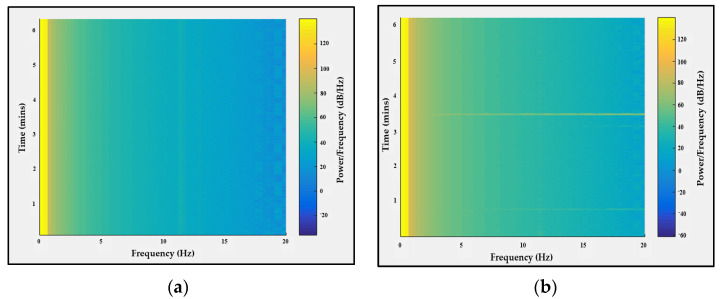
Alpha wave detection while (**a**) eyes were open and (**b**) eyes were closed.

**Figure 16 sensors-21-07703-f016:**
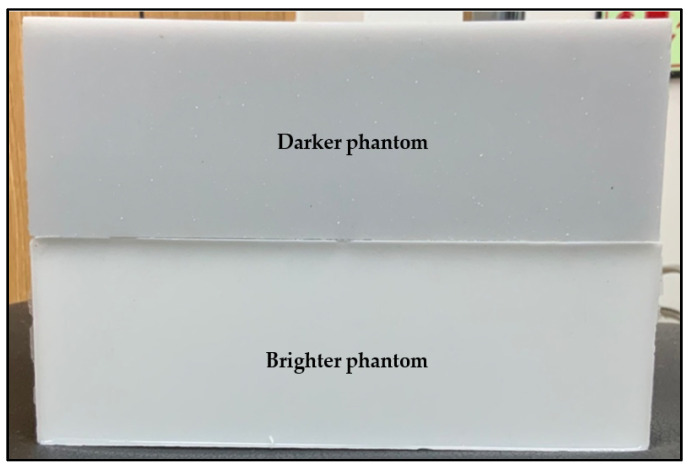
Darker and brighter silicone rubber phantoms.

**Figure 17 sensors-21-07703-f017:**
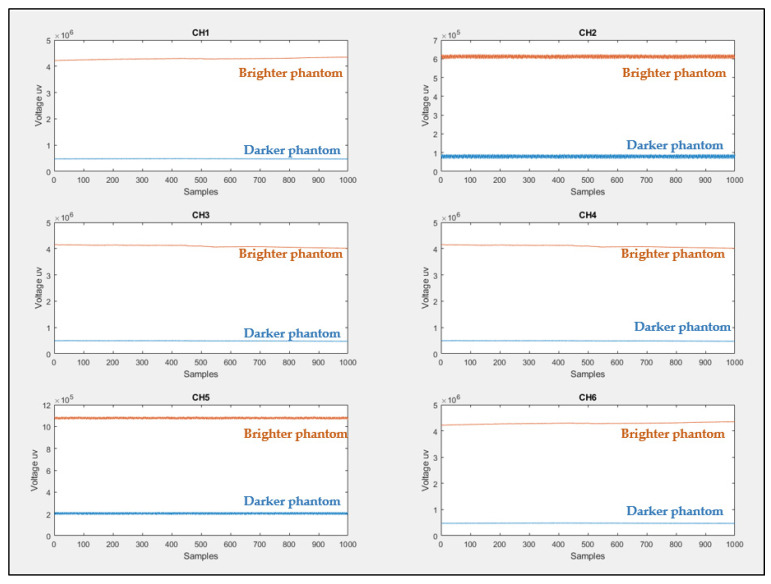
Light intensity sensed by PDs (six fNIRS channels) for darker and brighter silicone phantoms.

**Figure 18 sensors-21-07703-f018:**
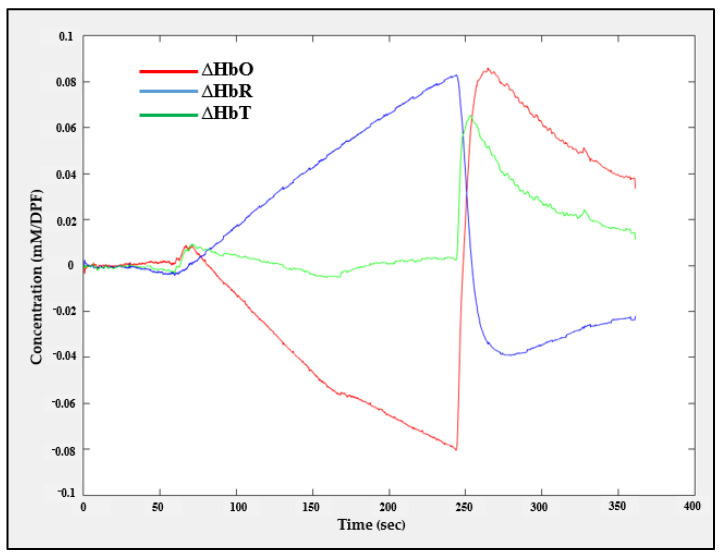
Hemodynamic responses recorded from channel one (CH1) during the arterial occlusion experiment.

**Figure 19 sensors-21-07703-f019:**
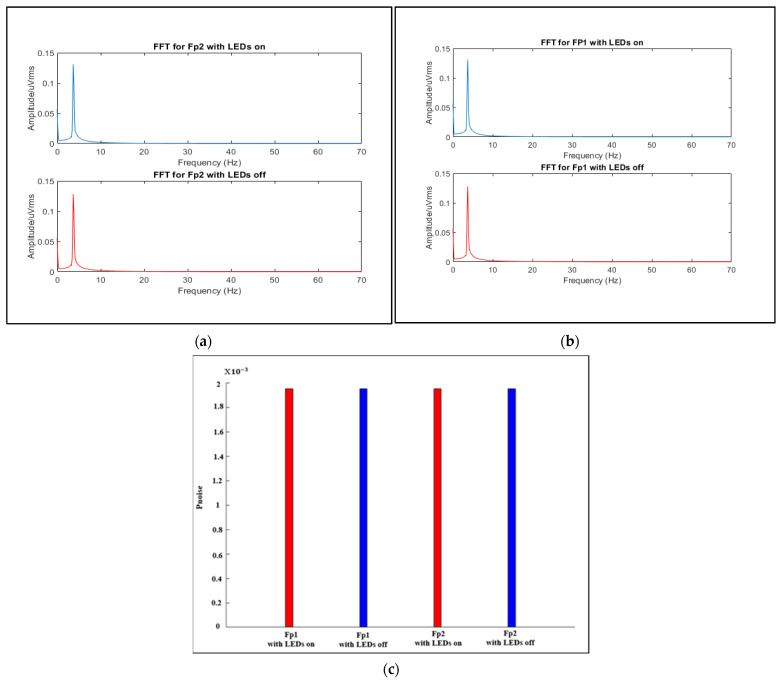
FFT spectra of signals measured at (**a**) Fp1 and (**b**) Fp2. (**c**) Sum of the extracted signal intensities at a 16 Hz fNIRS switching frequency.

**Figure 20 sensors-21-07703-f020:**
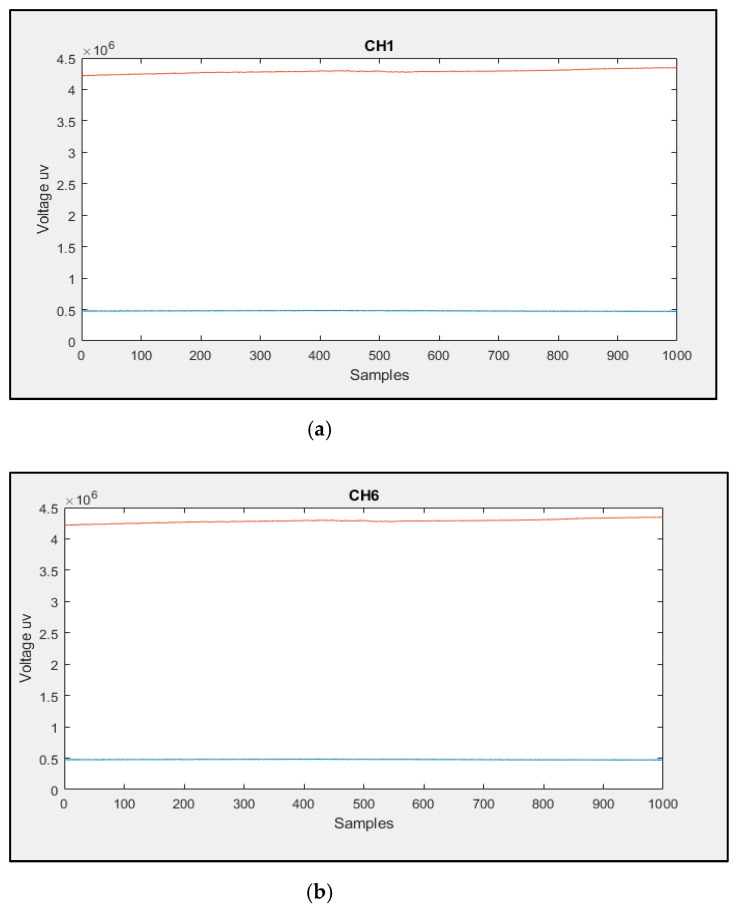
The recorded fNIRS data from (**a**) channel one and (**b**) channel six after subjecting the flexible PCB to 200 cycles of bending at different angles.

**Table 1 sensors-21-07703-t001:** EEG/fNIRS system cost.

No.	Item	Source	Unit Price (KRW)	Quantity	Total Price (KRW)
1	ADS1298IPAG	Mouser	33,500	1	33,500
2	TEENSY3.2	Mouser	32,000	1	32,000
3	BQ24380DSGT	Mouser	1200	1	1200
4	MCP73811/2	Mouser	13,000	1	13,000
5	TPS72301DBV T	Mouser	2600	1	2600
6	BQ29700DSET	Mouser	7400	1	7400
7	TPS73201DBV T	Mouser	7400	1	7400
8	LM2664M//NOPB	Mouser	1500	2	1500
9	Li-Polymer battery	Alibaba	2200	1	2200
10	2N3904	Mouser	95	4	380
11	Resistors, capacitors, etc.	Mouser	50,000		
	Total (KRW)		150,895		151,180

Note: the cost of PDs and LEDs is not included.

**Table 2 sensors-21-07703-t002:** Comparison of features between the proposed EEG/fNIRS system and seven previous EEG/fNIRS systems.

No.		Ref [9]	Ref [10]	Ref [11]	Ref [12]	Ref [13]	Ref [14]	Ref [15]	Current Study
1	Year of the study	2013	2013	2018	2019	2011	2017	2018	2021
2	Fabrication process	discrete	discrete	discrete	discrete	Microchip based	Microchip based	Microchip based	Discrete
3	# of EEG channels	8	8	32	16	-	2	1	2
4	EEG electrode position	10–10 standard	-	-	10–20 standard	-	AF7 and FT9	-	Fp1 and Fp2
5	EEG electrode type	Active wet	-	-	Active dry	-	Active dry	Active dry	Active wet
6	EEG electrode material	Ag/AgCl	-	-	-	-	-	-	Ag/AgCl
7	EEG resolution (bit)	16	16	16	24	10	12	15	24
8	EEG sampling rate (SPS)	1024	320	320	250	128	2000	-	250
9	CMMR (dB)	-	-	-	−110	-	>−110	−100	−115
10	number of sources	8	8	32	2	6	1	2	2
11	Number of detectors	4	8	32	6	12	1	2	5
12	number of fNIRS channels	32	32	128	8	24	1	4	6
13	LED wavelength/s (nm)	760,850	735,850	735,850	730,850	735,890	670,850	735,850	735,850
14	source-detector separation (mm)	20 to 63	31	30	27	14.14	-	30	30,5
15	fNIRS resolution (bit)	16	16	16	16	10	12	12	24
16	fNIRS sampling rate (SPS)	8	20	20	5	1	20–80	2–512	8
17	ADC setting for EEG and fNIRS	separated	shared	separated	separated	separated	shared	separated	Shared
18	Power consumption (mW)	400	2200	2600	18.8/ch	3.6 for chip	25.2	0.665 for chip	0.75/ch
19	Size of probe/cap	35 × 80 × 10 mm^3^	130 mm^3^	95 mm^3^	-	-	35 × 260 mm^2^	-	716 × 60 ^mm2^
20	Size of control unit	-	160 × 130 × 82 mm^3^	120 × 90 × 70 mm^3^	70 × 70 mm^2^ × 2	-	-	-	84 × 62 ^mm2^
21	system weight (g)	90	800	850	-	-	<26	-	73.5
22	System Cost (won)	-	-	-	-	-	-	-	151,180

## Data Availability

The datasets used and/or analyzed during the current study are available from the corresponding author on reasonable request.

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
