# Peer review of "Development of an Integrated EEG/fNIRS Brain Function Monitoring System"

_sensors, 2021, doi:10.3390/s21227703_

Round 1
Reviewer 1 Report
This manuscript described a development of an integrated EEG-fNIRS system.
There are significant amount of works are required for improvement.
First, the system proposed here didn’t demonstrate clear advantages over some existing systems. The sources and detectors numbers and channel numbers are very limited, and the source-detector separation is fixed, providing no depth info.
From a clinical point of view, this system will be extremely challenging to be used in practice for wearable monitoring. The wearability, patient comfort, stability to make a good scalp contact etc. remains questioning, and need significant analysis, measurement, and verifications.
Also, the authors suggested several age-related diseases, such as MCI, PD, AD, etc. in the introduction., I think none of these diseases can be effectively diagnosed by this current system, the logic and information is misleading, needs to be reconsidered.
In the introduction part, there are also some inaccurate information: for example the authors mentioned MEG, actually there is already a wearable MEG available developed by University of Nottingham, UK, published at Nature about 2 or 3 years ago; and some description about general fNRIS devices are also not fully correct: the sources of fNIRS not can only be LEDs, also other option available, like laser diode, laser, VCSELs etc., and the detectors can not only photodiodes, can also SiPM, APD, SPADs etc.
Meanwhile there are a lot key references are missed, for example:
- Kassab, A. et al.. Multichannel wearable fNIRS-EEG system for long-term clinical monitoring. Hum. Brain Mapp. 2018, 39, 7–23
- Chua, E et al.. A highly-integrated biomedical multiprocessor system for portable brain-heart monitoring. In Proceedings of the 2011 IEEE International Symposium of Circuits and Systems (ISCAS), Rio de Janeiro, Brazil, 15–18 May 2011
- Ha, U et al.. An EEG-NIRS Multimodal SoC for Accurate Anesthesia Depth Monitoring. IEEE J. Solid-State Circuits 2018, 53, 1830–1843.
- Xu, J et al.. A 665 μW Silicon Photomultiplier-Based NIRS/EEG/EIT Monitoring ASIC for Wearable Functional Brain Imaging. IEEE Trans. Biomed. Circuits Syst. 2018, 12, 1267–1277.
More relevant references are still required.
They also did compare this system with some commercial available systems such as Wearalbe Sensing, NIRx, Artinis, etc.
Some key parameters of the system are also unclear, such as the dark noise of the fNIRS measurement, the dynamic range of fNIRS measurement etc. More importantly, the potential crosstalk between EEG and fNIRS, which is a really important aspect, are not considered and investigated.
The overall writing of the manuscript can be more clear and scientific.
Reviewer 2 Report
See attached file.

Round 2
Reviewer 1 Report
Some concerns with current manuscript:
1) The robustness of flexible PCB approach remains questioning, is there detailed mechanical test and analysis conducted to verify the robustness and long-term stability?
2) the dynamic range of the fNIRS measurement, 74.45 dB, is quite low, the signal quality would be of concern, the measurement performance, particularly for haired-region would be unclear.
